# Death Anxiety and Attitudes towards Death in Patients with Multiple Sclerosis: An Exploratory Study

**DOI:** 10.3390/brainsci11080964

**Published:** 2021-07-22

**Authors:** Jara Francalancia, Paraskevi Mavrogiorgou, Georg Juckel, Tina Mitrovic, Jens Kuhle, Yvonne Naegelin, Ludwig Kappos, Pasquale Calabrese

**Affiliations:** 1Neuropsychology and Behavioral Neurology Unit, Division of Molecular and Cognitive Neuroscience, Department of Psychology, University of Basel, Birmannsgasse 8, 4055 Basel, Switzerland; j.francalancia@hotmail.com (J.F.); Tina.Mitrovic@unibas.ch (T.M.); 2Department of Psychiatry, Ruhr University Bochum, Alexandrinenstr. 1, 44791 Bochum, Germany; paraskevi.mavrogiorgou-juckel@lwl.org (P.M.); g.juckel@lwl.org (G.J.); 3Department of Neurology, University Clinic Basel, Petersgraben 4, 4031 Basel, Switzerland; Jens.Kuhle@usb.ch (J.K.); Yvonne.Naegelin@usb.ch (Y.N.); Ludwig.Kappos@usb.ch (L.K.)

**Keywords:** multiple sclerosis, anxiety, depression, fatigue, cognitive impairment, death anxiety, attitude towards death

## Abstract

Background: Death and the anxiety of it becomes more apparent when confronted with a chronic disease. Even though multiple sclerosis (MS) is a treatable condition today, it is still accompanied by a multitude of impairments, which in turn may intensify of death anxiety. Objective: The aim of this study is to explore the relationship between depression, anxiety and death anxiety in individuals with MS. Methods: Fifty-six MS patients were recruited at the Department of Neurology of the University Clinic in Basel. Death anxiety was assessed using the Bochumer Questionnaire on attitude to death and death anxiety 2.0 (BOFRETTA 2.0). Results: Scores of death anxiety towards it in MS patients were low. Only disability (EDSS) was moderately correlated with death anxiety. Depression in MS was significantly correlated with fatigue and disability, but not with the BOFRETTA 2.0. Conclusion: Scores of death anxiety and the attitude towards death are low in this MS cohort. It was shown that both psychopathological and neurological deficits impact the subject of death with respect to multiple sclerosis.

## 1. Introduction

The heterogeneous appearance of MS allows silent symptoms such as depression, anxiety and fatigue to fade into the background, since the physical impairments are primary in the treatment of MS. Today we know that clinically relevant depression and anxiety are more common in people with MS than in the general population, which significantly reduces their quality of life [1,2]. A systematic review [3] states the lifetime prevalence of clinically significant depressive symptoms in people with MS to be approximately 35%, others estimate it being even as high as 50% [4]. Somewhat lower are the prevalence values for anxiety disorders in MS patients, which vary from 22–36% [3,5].

It has been shown that there is increased death anxiety in patients with anxiety disorder [6]. Though there are differences in intensity and manifestation, the universality of concern for death is widely recognized and is an essential object of human thinking. While MS is a treatable condition today [7], the multitude of physical and psychological impairments that can arise throughout the course of the disease [3] may intensify awareness of and anxiety about death. Hence, it seems plausible that people who are confronted with a chronic disease that can severely restrict their lives are more likely to be engaged in thoughts and worries about death than healthy people [8]. Patients with MS have an average reduced life expectancy of 7–14 years, with range depending on the individual study [9,10,11]. Therefore, as the cause of death in MS patients in over two thirds is due to direct or indirect consequences of their disease [12], thoughts on existential questions may be more salient as well.

As the subject is already very sensitive for healthy people, it is not surprising that there is basically no research dealing with death anxiety in MS patients. Relevant to our question is the common wisdom that worsening health is stated to be associated with greater death anxiety [8]. Furthermore, increased death anxiety was found in patients with anxiety disorder [6]. Those results were found by Abdel-Khalek who repeatedly conducted studies on death anxiety in healthy and mentally ill patients [13]. Considering their findings, which state that up to 36% of MS patients have a clinically relevant anxiety disorder, the results from Abdel-Khalek offer some interesting possible insights.

Nonetheless, this area of clinical research still lacks substantial data. Therefore, the aim of this study is to illuminate the relationship between depression, anxiety and death anxiety in individuals with MS and the influence of the disease course and disability on these variables. The objective is to highlight the importance of affective disorders in chronically ill patients and the implications thereof.

## 2. Methods

### 2.1. Participants and Procedure

For this cross-sectional explorative study, a total of 56 MS patients (40 females; age: 50.8 ± 14.0 years; phenotype: 8 primary progressive, 40 relapsing remitting, 8 secondary progressive; disease duration: 14.9 ± 9.4 years) were recruited from the University Hospital of Basel by the end of February 2020. Participation was voluntary, and patients had to sign a written consent form prior to examination.

For the evaluation of death anxiety in particular we utilized the Bochum survey for assessment of attitude to death and death anxiety (BOFRETTA) [14] version 2.0, containing 22 statements about attitude towards death (11 items) and death anxiety (11 items). The scale is based on Templer’s death anxiety scale [15], Lester’s death anxiety scale and the question inventory for multi-dimensional evaluation of experience of dying and death (FIMEST). In the BOFRETTA scale, which is semi-quantitative, participants indicate their accordance with its statements by choosing between “does not apply at all” (1 point), “applies slightly” (2 points), “applies predominantly” (3 points) or “applies mostly” (4 points). As its qualitative component, participants may express personal thoughts or concerns towards death in three free-text columns. An example of this qualitative data concerning death anxiety: “Die Gewissheit, dass mein Leben auf der Erde begrenzt ist, macht mir Angst,” [The certainty that my life on earth is limited, scares me]; and an example concerning attitude: “Ich habe eine negative Einstellung zum Tod als das absolute Ende meines Lebens,” [I have a negative attitude to death as the absolute end of my life]. As further qualitative analysis, participants could express personal thoughts or concerns about death in three open questions. This questionnaire was sent to patients by mail after their individual sessions with an accompanying letter explaining the use of this questionnaire. The participants then returned the questionnaire with an enclosed reply envelope.

The demographic and clinical data of the MS patients were obtained from the patients’ medical records. Disability was also obtained from the patient’s medical records, since the examinations were done by a clinician during medical visits, employing the Expanded Disability Status Scale (EDSS) [16]. Patients were instructed and tested by a test evaluator in individual sessions. The tests and questionnaires used in the sessions covered cognition and fatigue, using the multiple sclerosis inventory of cognition (MUSIC), as well as depression and anxiety, using the hospital anxiety and depression scale (HADS [17]. MUSIC consists of five short subtests: a two-trial word-list-learning-paradigm (A1 and A2) and an additional one-trial distractor list (list B), a time-limited verbal fluency task with alternating categories (vegetables vs. clothes), a Stroop-like interference-test and, finally, a delayed recall measure. MUSIC raw-scores are weighted following discriminant analysis and transformed on the basis of mean and standard deviations. Age and education effects are considered by differential scoring (20–30 pts: *normal range;* 16–19 pts: *mild cognitive dysfunction*; 11–15 pts: *moderate cognitive dysfunction;* <10 pts: *severe cognitive dysfunction).*

### 2.2. Statistical Analyses

The data was analyzed with the statistics program Statistical Packages of Social Sciences (SPSS) [18]. Descriptive statistics were performed for every variable. To analyze correlations between the different variables, the bivariate correlation Spearman’s rho (r_s_) was used. Spearman’s partial correlation was used when controlled for potential covariables. To analyze group differences the Mann–Whitney U (z) test was used. The significance threshold was set at a *p*-value ≤ 0.05. All significant values were based on two-tailed tests.

## 3. Results

### 3.1. Descriptive Statistics

A total of 56 patients recruited from the University Hospital of Basel were included in the study. The sample consisted of 40 (71.4%) women and 16 (28.6%) men. The mean age was 50.75 (SD = 13.95) years, with a range from 27 to 77 years. On average, patients had suffered from MS for 14.911 years (SD = 9.389). Further, 8 (14.3%) patients were diagnosed with PPMS, 41 (73.2%) with RRMS and 7 (12.5%) with SPMS. Four (48.2%) out of 7 patients with SPMS were female, 8 (19.5%) out of 41 RRMS patients were male and 5 (62.5%) out of 8 PPMS female. The average age of PPMS patients was 61.1 years, SPMS were on average 48.1 years old and patients with RRMS 43.05 years old. The average disease duration in PPMS patients was 16.3 years, SPMS patients had an average duration of 19.3 years and RRMS 10, 1 years. The descriptive statistics can be seen in Table 1 and Table 2.

There were no significant correlations with sex, education or MS-Type. The results, sorted by phenotype (Table 3.), were as follows; Patients with PPMS showed an EDSS Score of 4.94 on average, RRMS had a Score of 2.47 and SPMS showed the highest impairment at 6.21. When assessed by means of the BOFRETTA 2.0 death anxiety subscale the average PPMS score was 11.5 (SD = 1.414). Considering the range from a minimum of 11 to a maximum of 44 points, this is a mean score within the lower quarter. The scores of SPMS patients were similar, with a BOFRETTA score of 12.86 (SD = 2.410). Only RRMS patients were in the upper lower quarter, at 14.59 (SD = 4.111). In terms of anxiety, SPMS patients showed the highest values with 4.43 (SD = 2.820) whereas RRMS and PPMS patients both had average scores below 4. The same applies to depression symptoms, which were highest in SPMS patients with a mean value of 5, whereas RRMS and PPMS Patients also had a value below 4.

### 3.2. Psycho-Affective Comorbidities

#### 3.2.1. Anxiety Symptoms

For the HADS-A, the average patient score was 3.66 (SD = 3.225), falling within the non-anxious range (HADS-A score ≥ 8). The majority (47) of patients (83.9%) showed no signs of anxiety, while six (10.7%) were in the range of a possible clinically significant anxiety and three (5.4%) in the range of definitely clinically relevant anxiety. Between anxiety and depression, there was a strong correlation, even after controlling for the EDSS score and other covariates (rs(56) = 0.67, *p* < 0.001). Furthermore, there was a moderate correlation between anxiety and attitude towards death (rs(56) = 0.28, *p* < 0.05), but a strong one with death anxiety (rs(56) = 0.40, *p* < 0.01). However, after controlling for depression this association was no longer significant for attitude towards death (rs(56) = 0.25, *p* = 0.068). For death anxiety (rs(56) = 0.28, *p* = 0.038) and the sum score (rs(56) = 0.30, *p* = 0.028) correlations remained significant. A moderate to strong correlation was found between anxiety symptoms and the fatigue score (rs(56) = 0.45, *p* < 0.01). No significant relationships were found regarding age (rs(56) = 0.16, *p* = 0.226), CI (rs(56) = −0.17, *p* = 0.134) and EDSS score (rs(56) = 126, *p* = 0.355). Finally, there was no significant gendered difference (z = −0.773, *p* = 0.440).

#### 3.2.2. Depressive Symptoms

The mean HADS-D score was 3.07 (SD = 3.324), indicating a score which was within the non-depressive range (HADS-D score ≥ 8). Of the patients interviewed, 49 (87.5%) were in the non-depressive range, according to the HADS, while 5 (8.9%) scored in the possible range and just 2 (3.6%) patient scores were definitely in the range of clinically relevant depression symptoms. There were no significant correlations between depression and attitude towards death (rs(56) = 0.05, *p* = 0.722), death anxiety (rs(56) = 0.24, *p* = 0.070) or the sum score (rs(56) = 0.13, *p* = 0.328). However, after controlling for anxiety, this association remained insignificant for attitude towards death (rs(56) = 0.056, *p* = 0.687), death anxiety (rs (56) = 0.21, *p* = 0.125) or sum score (rs(56) = 0.15, *p* = 0.278). A strong correlation was found between depressive symptoms and the fatigue score (rs(56) = 0.69, *p* < 0.001), CI (rs(56) = −0.29, *p* < 0.05) and EDSS score (rs(56) = 0.45, *p* < 0.001). No significant relationship was found with age (rs(56) = 0.15, *p* = 0.281). There was also no significant gendered difference (z = −0.046, *p* = 0.936) (see Table 4).

### 3.3. Death Anxiety and Attitude towards Death

When assessed by means of the BOFRETTA 2.0, the death anxiety subscale the average patient score was 13.93 (SD = 3.808). Considering the range, from a minimum of 11 to a maximum of 44 points, this is a mean score within the lower quarter. Moreover, the highest score obtained from any patient interviewed was 30. Similar patterns apply to BOFRETTA 2.0 attitudes toward the death subscale, with the same possible range of 11–44, where we found a mean patient score of 19.38 (SD = 3.691), and thus not a particularly negative one; the highest score obtained was 31. The sum score had a mean of 33.3 (SD = 6.761) with a range from 22 to 61. In total, 55 (98.2%) patients had a score of 47 or lower, meaning that most of the patients had a relatively low total value and only one (1.8%) patient scored comparatively high (see Table 5). Furthermore, there was a moderate correlation between anxiety (HADS-A) and attitude towards death (rs(56) = 0.28, *p* < 0.05) and a strong one between death anxiety (rs(56) = 0.40, *p* < 0.01) and sum scores (rs(56) = 0.37, *p* < 0.01). In comparison, there was no significant correlation between depression (HADS-D) and attitude towards death (rs(56) = 0.05, *p* = 0.722), death anxiety (rs(56) = 0.24, *p* = 0.070) or sum score (rs(56) = 0.13, *p* = 0.328). On the one hand, significant correlations between the BOFRETTA 2.0 and the HADS-D disappeared when controlling for anxiety. With HADS-A, on the other hand, there were no significant correlations with attitude towards death (rs(56) = 0.06, *p* = 0.687), death anxiety (rs (56) = 0.21, *p* = 0.125) and sum score (rs(56) = 0.15, *p* = 0.278), even after controlling for depression. Looking into the group differences between anxious and non-anxious people (HADS-A scores ≥ 8), significant group differences could be found in death anxiety (z = −3.24, *p* = 0.001, |d| = 0.43), attitude towards death (z = −0.04, *p* = 0.041, |d| = 0.27) and sum score (z = −2.93, *p* = 0.003, |d| = 0.39). The same applied to depressive and non-depressive people (HADS-D scores ≥ 8) regarding death anxiety (z = −3.01, *p* = 0.003, |d| = 0.4), attitude towards death (z = −2.51, *p* = 0.012, |d| = 0.33) and sum score (z = −2.29, *p* = 0.004, |d| = 0.38). All effects except for one were moderate with respect to size (Cohen, 1992). No significant correlation could be identified with death anxiety, fatigue (rs (56) = 0.114, *p* = 0.403) and CI (rs(56) = 0.04, *p* = 0.754). Only EDSS score showed significant correlation with death anxiety (rs(56) = −0.29, *p* = 0.05) Attitude towards death did not correlate significantly with fatigue (rs(56) = −0.09, *p* = 0.495), CI (rs(56) = −0.10, *p* = 0.469) and EDSS scores (rs(56) = −0.172, *p* = 0.204). Sum score showed no significant correlation with fatigue (rs(56) = −0.003, *p* = 0.983) or CI (rs(56) = −0.005, *p* = 0.970). As with death anxiety, sum scores only correlated significantly with EDSS score (rs(56) = −0.26, *p* < 0.05). When controlling for CI and fatigue, the relationships between death anxiety and EDSS score (rs(56) = −0.27, *p* = 0.043) and sum score (rs(56) = −0.27, *p* = 0.043) remained significant.

### 3.4. Fatigue and EDSS-Score

According to the fatigue scale of the MUSIC, the mean value was 9.93 (SD = 5.68), which is above the cut off for a clinically relevant fatigue. A total of 32 (55.2%) patients showed no clinically relevant fatigue, whereas 24 (41.5%) had a score that indicated clinically relevant fatigue. There were two (3.4%) missing values. As can be seen from Table 6, a moderate to strong correlation was found between fatigue score and anxiety symptoms (rs(56) = 0.45, *p* < 0.01) and depressive symptoms (rs(56) = 0.69, *p* < 0.01). No significant correlations were found with all three subscales of the BOFRETTA 2.0. The mean EDSS score was 3.438 (SD = 1.851), which indicated moderate disability in one functional system or mild disability in three to four functional systems with full walking ability, with a range of 0–7. There was a moderately significant correlation between depressive symptoms and EDSS score (rs(56) = 0.45, *p* < 0.001). However, no correlation was found between EDSS score and anxiety symptoms (rs(56) = 0.126, *p* = 0.355). There were significant correlations with death anxiety (rs(56) = −0.29, *p* < 0.05) and sum score (rs(56) = −0.26, *p* < 0.05). There had been a moderate significant correlation between fatigue and EDSS score (rs(56) = 0.43, *p* < 0.01), but was no longer significant after controlling for CI (rs(56) = 0.23, *p* = 0.086). Both showed significant correlations with CI; fatigue showed a moderately significant correlation (rs(56) = −0.39, *p* < 0.01) and EDSS score showed a strongly significant correlation (rs(56) = −0.51, *p* < 0.001). Fatigue and CI no longer correlated significantly after controlling for EDSS score (rs = −0.23, *p* = 0.087). After controlling for fatigue, CI and EDSS score still correlated significantly. Contrary to fatigue (rs(56) = 0.16, *p* = 0.254), EDSS score showed a significant correlation with age (rs(56) = 0.62, *p* < 0.001).

To assess general CI, the standardized test MUSIC was used. The mean value for MUSIC was 22.32 (SD = 6.247) which indicated no CI. A total of 37 (66.1%) of the patients had a score that indicated no CI, while 6 (10.7%) had an indication of mild, 12 (21.4%) moderate and only 1 (1.8%) heavy CI, while 6 (10.7%) had indications of mild CI, 12 (21.4%) of moderate CA and only 1 (1.8%) of severe CI. A moderate correlation was found between CI (rs(56) = −0.29, *p* < 0.05) and depressive symptoms. A moderate significant correlation (rs(56) = −0.39, *p* < 0.01) was found with CI and fatigue. Finally, a significant correlation between CI and the EDSS score could be seen (rs(56) = −0.51, *p* < 0.001). After controlling for fatigue, CI and the EDSS score still correlated significantly. The partial correlation between CI and fatigue was no longer significant, after controlling for the EDSS score (rs = −0.23, *p* = 0.087). Finally, CI did not correlate significantly with any subscale of the BOFRETTA 2.0.

## 4. Discussion

The aim of this study was to explore the relationship between depression, anxiety and death anxiety in individuals with MS. It aimed to investigate possible relationships between psycho-affective comorbidities (i.e., depression and anxiety) and death anxiety in a Swiss population of MS patients, in particular what role the impairments here considered (EDSS score, fatigue, CI) play in the experience of death anxiety.

The interesting findings of this study are that both anxiety (16.1%) and depression values (12.5%) were below the estimated prevalence values for MS patients in the literature [3,5]. Furthermore, no gendered differences were found. Regarding their general impairment, the sample showed clinically significant fatigue in the mean (*M* = 9.93) with a ceiling of 21. Regarding CI, the sample, on average, scored none. Only 32.1% showed mild to moderate CI and only one scored severe.

Looking at the correlations between the HADS-A and all the subscales of the BOFRETTA 2.0 [19] it is noteworthy that all correlations were significant. Interestingly, after controlling for depression, associations remained significant only for death anxiety and sum score. Additionally, there was a moderate to strong correlation between HADS-A and fatigue. No significant correlation was found between anxiety, age, MUSIC or EDSS. In other words, anxiety was not significantly associated with age, CI or disability in MS patients. However, there were significant group differences between anxious and non-anxious patients. Patients with anxiety symptoms had clinically relevant fatigue symptoms more often than those without. This may be due to reduced activity and ability to participate in every-day life, since fatigue symptoms include the exhaustion of reserves of energy or increased need for rest that is disproportionate to recent effort [20].

With consideration to clinical and demographic variables, no correlations were found between HADS-D and all the subscales of BOFRETTA 2.0. Contrary to HADS-A, HADS-D showed significant correlations with MUSIC, fatigue scores or EDSS, meaning that depressive symptoms are more likely to be associated with fatigue symptoms, disability, and CI. As symptoms of fatigue often overlap with symptoms of depression (e.g., exhaustion, tiredness, etc.), the strong relationship between depression and fatigue makes sense. In fact, some studies have found that there is a correlation between fatigue and depression [21,22]. Furthermore, the connection between depression and disability (EDSS) is also comprehensible, as both can be traced back to structural changes in the central nervous system [23]. Depression could therefore be partly explained by disability and fatigue. The same applies, partially, to CI. Numerous studies have shown that CI has a negative impact on other parts of life, such as social and working life, which can lead to unemployment and low quality of life [24,25]. Factors such as unemployment and quality of life are, in turn, also associated with depression [26].

Looking at the different correlations with the other factors, one is particularly interesting: only correlations with the HADS-A were significant, even after controlling for depression but not vice-versa. However, both group differences, between anxious and non-anxious as well as depressive and non-depressive patients, remained significant. Additionally, all effect sizes, except for attitude towards death and anxious/non-anxious patients, showed a moderate effect. This could be seen as the patients converting their death anxiety into “situational anxiety”, which is seen as a coping strategy, since it limits the perception of one’s own finitude [27]. In comparing these findings to the general impairment of MS patients no correlations were found. Neither fatigue nor cognitive impairment showed significant correlations with BOFRETTA 2.0. Only EDSS score correlated significantly with either death anxiety or sum score after controlling for covariables. This can be interpreted as different measures for impairment being overlapping constructs. Since EDSS score considers cognitive domains as well as physical limitations it is a good measure for obtaining a general impression of a patient’s impairment.

Examining the results, it is apparent that the sample, in general, had low scores in all three subscales of the BOFRETTA 2.0. The mean score in the death anxiety subscale (M = 13.93) was not even half the maximum possible score. Attitude towards death scored just a bit higher, on average (M = 19.38), as it did for sum scores (M = 33.3) of BOFRETTA 2.0. Most strikingly, 98.2% of the participants had a score of 47 or lower. As rates of death anxiety are not high in MS patients in general, according to our findings, one might ascribe this to the fact that views about death are very individual. The fact that one has health issues or a diagnosis like MS does not necessarily result death anxiety, as the individual might have a positive attitude towards it. Taking the findings of Juckel and Mavrogiorgou [14] and Chmielewski et al. [19] into account, personality traits could explain some of the results. In all previous BOFRETTA studies a positive relationship between neuroticism and death anxiety was found. This means that the low scores in death anxiety, in this sample, could be discussed as a manifestation of a personality dimension. It is possible that this sample was low on neuroticism or had an above-average increased openness, both of which, in turn, may be preventive against anxiety. Another reason for this low score may be the relatively long disease duration of each patient. Since MS is a life-long disorder and the average disease duration in this sample was 15 years, it could be argued that early or enduring confrontation with a disabling disease has led to a more mature attitude towards human finitude.

The challenge in research on death anxiety is definition. First, from within the different types of established death anxiety, our focus was on the personal prospects of death and dying; that is, existential death anxiety, which stems simply from the knowledge that human life is finite. Secondly, the capability for realizing this finitude is bound to different conditions, such as awareness of the distinction between self and other, fullness of sense of personal identity, and ability to anticipate the future [28]. Hence, we take death anxiety as the “possibility of a mental occupation with dying and death”, “which is associated with anxiety, tension, dysphoric excitement and the like” and assume it to reflect knowledge of one’s own and others’ mortality [29]. This awareness can lead to fearing the inevitable, i.e., death anxiety. There are many definitions and theories around death anxiety, some of them distinguish between fear “of a thing” and fear of “something that is nothing” and some others describe death anxiety as “a negative emotional reaction provoked by the anticipation of a state in which the self does not exist” [30]. Hoffmann [31] differentiates between reactive and existential fears of death. Reactive fear implies a specific stressful situation, which subsides again after the duration of the acute stress phase or else is localized externally, thus giving the individual the opportunity to adapt to the fear. In this context, fear of death can be interpreted as a state of fear “that accompanies a subjectively experienced threat to one’s own life” and as such, is a natural reaction to acute danger to life. An existential fear, on the other hand, is more prolonged and hence corresponds more to a “basic emotional state”. In this case, the fear is located within the person and cannot be confronted, thus is not opposable. It seems apparent that in order to reduce existential fear, many people will subconsciously transform it into situational fear. This conversion of death anxiety into situational anxiety limits the perception of one’s own finitude and has therefore been regarded as a coping strategy [27]. Failure of such coping strategies may serve as a catalyst for the establishment of mental disorders like depression, as the pathologic dimensions of death anxiety have been considered to relate to other mental disorders, e.g., anxiety disorders [32]. Interestingly, although evidence indicates that death anxiety is expected to be the most powerful unconscious psychodynamic dynamism in today’s emotional life [33], it remains a taboo topic that lacks substantial data.

With respect to our study and considering the low BOFRETTA scores obtained, the impression could arise that a substantial fraction of our participants did not have strong feelings, anxieties or attitudes toward death. This might be due to the fact that MS is a treatable disease, today, and participants were not in a constant state of concern for their death since their disease’s chronicity might induce adaptation with time. Interestingly, and somewhat contrary to our assumptions, chronic disease itself was not responsible for increased levels of death anxiety. It is more likely that personal dispositions, comorbidities and affective disorders, rather than physical impairment, play a role here. Essentially, this is also what was shown in the results: the actual psychopathological status (representing an anxiety-depressive mood) has a much greater impact on the subject of death than the limitations caused by the neurological disorder itself. This is similar to the results obtained from patients with amyotrophic lateral sclerosis, who showed relatively low levels of death anxiety despite being even more impaired than MS patients [34]. Nevertheless, the psychopathology must be seen in the context of the physical illness since the physical impairments correlate relatively highly with depression and anxiety. Finally, however, it can be concluded that attention should be paid to an anxiety comorbidity in MS patients in order to treat this appropriately with psychiatric-psychotherapeutic treatment.

### Conclusions, Limitations and Further Prospects

This study aimed to (1) investigate possible relationships between psycho-affective comorbidities (i.e., depression and anxiety) and death anxiety in MS patients in a Swiss population and, in particular, (2) investigate the role of impairments (EDSS score, fatigue, CI) play in experienced death anxiety. Finally, (3) the study sought to draw from the qualitative data of the BOFRETTA 2.0 when discussing quantitative results. A significant correlation between anxiety and depression was found, as well as significant correlations between anxiety and fatigue, depression and fatigue, CI and EDSS score. These findings were consistent with other studies. However, this study revealed a lower prevalence of depression and anxiety than previous studies. Nevertheless, the most important results of this study refer to the newly developed BOFRETTA 2.0 questionnaire. It is the first time this questionnaire has been used on an MS cohort and the first time the issue of fear of death has been addressed in this way. Contrary to expectations, fear of death and the attitude towards it was low in this MS cohort. It was found that current psychopathology has a much stronger impact on the subject of death than the limitations caused by their neurological disease. Except with EDSS score, there were no significant correlations of impairment with BOFRETTA 2.0. Finally, even a burdensome diagnosis like MS does not necessarily lead to fear of death or depression and can be seen by many patients as an opportunity to make the best of it. Future research should certainly take a closer look at the subject of death because, especially in open questioning, many interesting aspects came to light that could not be dealt with here. Furthermore, it would be of great interest to study the issue of death attitudes as extensively as possible in different neurodegenerative cohorts, since little or no research has been done on the subject in the literature to date. There is a lot of potential in this topic, which will not only benefit those directly affected but also humanity in general, since it is innately bound to that which affects us all equally: the realities of sickness and of health.

The limitations of this study include a rather small (N = 56) sample size and the lack of a control group. Additionally, phenotypes were not equally represented in the sample and the SPMS and PPMS subtypes did not even account for 30% of the sample. Unfortunately, a lot of possible participants were lost because of our—possibly provocative–questionnaire: only 56 of 110 issued questionnaires were returned. This might have led to a selection bias, since we infer only patients who were not triggered by the topic or were not severely concerned about the subject returned it. Furthermore, we have not controlled our results for disease-modifying therapies or annual relapse-rates (ARR), which might have had influence on the results. It is also of importance to note that our data were collected shortly before the COVID-19 pandemic which also might have led to some relevant answering tendencies of participants.

Ultimately, to assess psycho-affective symptoms as well as general CI and fatigue, only screening tools were used. These screening tools do not allow an in-depth diagnosis and only serve to preselect patients with possible disorders or impairments who might need a more detailed assessment. Future research should certainly take a closer look at the subject of death, because, especially as an open question, many interesting aspects emerged that were beyond the current scope. Furthermore, it would be of great interest to study the issue of death as extensively as possible in different neurodegenerative cohorts since little or no research has been done on the subject in the literature to date.

## Figures and Tables

**Table 1 brainsci-11-00964-t001:** Descriptive Statistics of Demographic and Clinical Variables (N = 56).

Variable	Mean (SD)	Median	Range
age (years)	50.75 (±13.951)	51	27–77
education (years)	15.39 (±4.355)	15	8–37
disease duration (years)	14.91 (±9.39)	12	3–42
MUSIC (Fatigue)	9.93 (±5.679)	9.5	3–21
MUSIC (CI)	22.32 (±6.247)	23	10–32
EDSS-Score	3.44 (±1.851)	3.3	0–7
HADS-A	3.66 (±3.255)	3.0	0–12
HADS-D	3.07 (±3.324)	2.0	0–13
BOF_ANX	13.93 (±3.808)	12.0	11–30
BOF_ATT	19.38 (±3.691)	19.0	11–31
BOF_SUM	33.34 (±6.761)	31.0	22–62

Note. Cognitive impairment and fatigue were measured with the multiple sclerosis inventory of cognition (MUSIC).

**Table 2 brainsci-11-00964-t002:** Frequencies of Demographic and Clinical Variables (N = 56).

Variable		Frequencies	
		n	%
sex	female	40	71.4
	male	16	28.6
fatigue	no fatigue	32	57.1
	fatigue	24	42.9
MS phenotype	RRMS	41	73.5
	PPMS	8	14.3
	SPMS	7	12.5
cognitive impairment	no	37	66.1
	mild CI	6	10.7
	moderate CI	12	21.4
	severe CI	1	1.8
anxiety	no	47	83.9
	possible anxiety	6	10.7
	anxiety	3	5.4
depression	no	49	87.5
	possible depression	5	8.9
	Depression	2	3.6

Note. RRMS = relapsing-remitting Multiple Sclerosis. PPMS = primary progressive Multiple Sclerosis. SPMS = secondary progressive Multiple Sclerosis.

**Table 3 brainsci-11-00964-t003:** Descriptive Statistics of Demographic and Clinical Variables sorted by phenotype (N = 56).

Phenotype (n)	Variable	Mean (SD)
PPMS (8)	EDSS	4.94 (±1.116)
	BOF_ATT	16.88 (±3.720)
	BOF_ANX	11.5 (±1.414)
	BOF_SUM	28.38 (±4.879)
	HADS-A	2.25 (±3.151)
	HADS-D	3.50 (±3.338)
	fatigue	10.63 (±3.998)
	MUSIC	21.13 (±6.058)
RRMS (41)	EDSS	2.47 (±1.244)
	BOF_ATT	20.24 (±3.419)
	BOF_ANX	14.59 (±4.111)
	BOF_SUM	34.83 (±6.749)
	HADS-A	3.80 (±3.333)
	HADS-D	2.66 (±3.299)
	fatigue	9.37 (±6.082)
	MUSIC	23.22 (±6.114)
SPMS (7)	EDSS	6.21 (±0.488)
	BOF_ATT	17.14 (±3.388)
	BOF_ANX	12.86 (±2.410)
	BOF_SUM	30.00 (±5.132)
	HADS-A	4.43 (±2.820)
	HADS-D	5.00 (±3.162)
	fatigue	12.43 (±4.504)
	MUSIC	18.43 (±6.373)

Note. RRMS = relapsing-remitting Multiple Sclerosis. PPMS = primary progressive Multiple Sclerosis. SPMS = secondary progressive Multiple Sclerosis.

**Table 4 brainsci-11-00964-t004:** Spearman bivariate correlation and partial correlation of the BOFRETTA 2.0 with group differences between anxious and non-anxious as well as depressive and non-depressive patients (N = 56).

		Correlation		Adjusted Correlation ^a^	
		rs	*p*	rs	*p*
HADS-A	HADS-D	0.668 **	<0.001		
	BOF_ANX	0.278 *	<0.05	0.381 *	0.038
	BOF_ATT	0.395 **	<0.01	0.284	0.068
	BOF_SUM	0.373 **	<0.01	0.297 *	0.028
	EDSS	0.126	0.355		
	fatigue	0.447 **	<0.01		
	CI	−0.176	0.194		
HADS-D	HADS-A	0.668 **	<0.001		
	BOF_ANX	0.244	0.070	0.209	0.125
	BOF_ATT	0.049	0.722	0.056	0.687
	BOF_SUM	0.133	0.328	0.149	0.278
	EDSS	0.452 **	<0.001		
	Fatigue	0.692 **	<0.001		
	CI	−0.291 *	<0.05		

Note: ^a^ Adjusted for covariates; ** significant with *p* < 0.01; * significant with *p* < 0.05.

**Table 5 brainsci-11-00964-t005:** Spearman bivariate correlation and partial correlation of the HADS (N = 56).

		Correlation		Adjusted Correlation ^a^		Group Differences		
		*rs*	*p*	*rs*	*p*	*z*	*p*	*|d|*
BOF_ANX	HADS-A	0.3950 **	<0.01			−3.241 *	0.001	0.433
	HADS-D	0.244	<0.01			−3.010 *	0.003	0.402
	fatigue	0.114	0.403	0.253	0.065			
	EDSS	−0.294 *	<0.05	−0.276 *	0.043			
	CI	0.043	0.754	−0.003	0.985			
BOF_ATT	HADS-A	0.278 *	<0.05			−2.043 *	0.041	0.273
	HADS-D	0.049	0.722			−2.506 *	0.012	0.334
	fatigue	−0.093	0.495	0.004	0.977			
	EDSS	−0.175	0.204	−0.225	0.102			
	CI	−0.099	0.469	−0.137	0.325			
BOF_SUM	HADS-A	0.373 **	<0.01			−2.930 *	0.003	0.391
	HADS-D	0.133	0.328			−2.857 *	0.004	0.381
	fatigue	−0.003	0.983	0.145	0.294			
	EDSS	−0.264 *	0.049	−0.276 *	0.043			
	CI	−0.0059	0.970	−0.078	0.577			

Note. ^a^ Adjusted for covariates; ** significant with *p* < 0.01; * significant with *p* < 0.05.

**Table 6 brainsci-11-00964-t006:** Spearman’s bivariate correlation and partial correlation of the three different impairment measures: fatigue (MUSIC), disability (EDSS) and cognitive impairment (MUSIC) with the BOFRETTA 2.0 and the HADS (N = 56).

		Correlation		Adjusted Correlation ^a^	
		*rs*	*p*	*rs*	*p*
fatigue	HADS-A	0.447 **	<0.01		
	HADS-D	0.692 **	<0.001		
	BOF_ANX	0.114	0.403	0.253	0.065
	BOF_ATT	−0.093	0.495	0.004	0.977
	BOF_SUM	−0.003	0.983	0.145	0.294
EDSS score	HADS-A	0.126	0.355		
	HADS-D	0.452 **	<0.001		
	BOF_ANX	−0.294 *	<0.05	−0.276 *	0.043
	BOF_ATT	−0.175	0.204	−0.225	0.102
	BOF_SUM	−0.264 *	0.049	−0.276 *	0.043
CI	HADS-A	0.373 **	<0.01		
	HADS-D	−0.291 *	<0.05		
	BOF_ANX	0.043	0.754	−0.003	0.985
	BOF_ATT	−0.099	0.469	−0.137	0.325
	BOF_SUM	−0.0059	0.970	−0.078	0.577

Note. ^a^ Adjusted for covariates; ** significant with *p* < 0.01; * significant with *p* < 0.

## Data Availability

The data presented in this study are available on request from the corresponding author. The data are not publicly available due to restrictions e.g., privacy or ethical.

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
