# Peer review of "Death Anxiety and Attitudes towards Death in Patients with Multiple Sclerosis: An Exploratory Study"

_brainsci, 2021, doi:10.3390/brainsci11080964_

Round 1

Reviewer 1 Report

well explained and discussed exploratory study with a small sample size considering the chosen methodology. I can note an imbalance when we consider MS phenotype: I suggest the authors to indicate data in Table 1 separately according to RMS or PMS. Switching therapy or high ARR in active patients could be affect results in the area of anxiety, please add these data if disposable.  Disease duration is a fundamental point, as discussed in the paper, and it cold be useful emphasize eventual deviations in the relapsing group, if present, also according  to cognitive involvement. Collection data is previous Covid pandemic break out, so spontaneously I wonder if nowadays the result would be the same...if possible, might be interesting to contact again the patients to retest them with the BOFRETTA scale or, alternatively, to consider this point in the prospect for further research at least. In case, to consider and update references with paper on coping strategies evaluated during pandemic on MS population.

Author Response

Reviewer 1:

well explained and discussed exploratory study with a small sample size considering the chosen methodology. I can note an imbalance when we consider MS phenotype: I suggest the authors to indicate data in Table 1 separately according to RMS or PMS.

We thank the reviewer for the positive attitude towards our paper. We have changed the table accordingly by adding the values for the different MS-phenotypes. Table 2 contains now the clinical data, showing that the RRMS-subgroup containing the most participants, which reflects clinic-epidemiological reality.

Switching therapy or high ARR in active patients could be affect results in the area of anxiety, please add these data if disposable. Disease duration is a fundamental point, as discussed in the paper, and it could be useful emphasize eventual deviations in the relapsing group, if present, also according to cognitive involvement.

We thank the reviewer for her/his constructive recommendations; though we have not collected the ARR at hand, we have specified that our population was clinically stable at the time of testing. Moreover, we have again taken this point in the limitations section by discussing possible its influence.

Collection data is previous Covid pandemic break out, so spontaneously I wonder if nowadays the result would be the same...if possible, might be interesting to contact again the patients to retest them with the BOFRETTA scale or, alternatively, to consider this point in the prospect for further research at least.

We thank the reviewer for this critical point which we lso have now mentioned in the restructured discussion section.

Reviewer 2 Report

In this study, Francalancia J. and coauthors explored death anxiety and attitude towards death in a population of patients with multiple sclerosis. The topic is of great interest, since it seems plausible that patients with MS are more engaged in thoughts and worries about death than the healthy population, potentially resulting in a decreased quality of life. Moreover, study evaluating novel tools to assess psychological impairment in MS are urgently warranted. However, the present study presents several limitations in this regard. Indeed, the BOFRETTA 2.0 scale, which represents the core evaluation of the study, is not sufficiently explored, as the title of the manuscript would suggest. Following my major comments:

-Since the BOFRETTA 2.0 scale is not extensively used in the MS field and many neurologists may be unfamiliar with it, I would suggest the authors to describe more in detail the BOFRETTA 2.0 scale, providing (i) some examples of questions about death anxiety AND attitude towards death, (ii) describing in more detail how the sub-scores are obtained and (iii) reporting the normative values of this test. If, as I might understand, no normative values for this scale yet exist, I would suggest the authors to describe in more detail the mean-median-standard deviation-interquartile range of these scores reported by other authors in other populations. The reference 19, which is reported as normative control group, is in German and not easily accessible to international readers.

-The authors did not explore the impact of the MS phenotype on the numerous test scores reported in this study. I would suggest to report the test scores both in the whole patient population AND for RRMS and progressive MS separately. The authors should systematically explore between group differences in these two patient groups. Similarly, I suggest to explore correlations between the several NPS scores with disease duration, age and education and to explore the effect of sex.

-The authors should explicit how cognitive impairment was defined based on the Multiple Sclerosis Inventory of Cognition scores ( how "mild", "moderate", "heavy" cognitive impairment was defined?). The authors should as well report the cut-off values they used for HADS-A and HADS-D for defining "possible" anxiety/depression, presence and absence of anxiety/depression.

-Table 3. It is not clear to me which groups of patients were compared and why these group differences have not been explored systematically for all patient groups (depressed/not depressed, fatigue/no-fatigue, cognitive impaired/cognitively preserved, death anxiety-no death anxiety...). Moreover, group differences are not reported and discussed in the text. I suggest to reduce correlation analyses in the results section (bivariate and partial correlation analyses could be reported in a single table: authors can report only more meaningful results in the text thus making the reading more clear). 

-The authors report as a limitation of the study the absence of data on the years of education to correct for this confounder, but they report it in table 1.

-In the Discussion, the authors do not concentrate on death anxiety and attitude towards death assessed by BOFRETTA 2.0 (which should be the main topic of the paper) but rather discuss associations between depression/fatigue/cognitive impairment in MS patients, which are well established in the MS literature. The novelty of this work, which is to investigate death anxiety and to introduce a new NPS tool, is thus not sufficiently developed.

-Abstract: the statement "It was shown that current psychopathology has a much stronger impact on the subject of death than the limitations caused by the neurological disease" is too declarative and it is not supported by the results of the study. Indeed EDSS is correlated with death anxiety. The term "current psychopathology" is too vague (what do the authors want to suggest?) 

Author Response

Reviewer 2

In this study, Francalancia J. and coauthors explored death anxiety and attitude towards death in a population of patients with multiple sclerosis. The topic is of great interest, since it seems plausible that patients with MS are more engaged in thoughts and worries about death than the healthy population, potentially resulting in a decreased quality of life. Moreover, study evaluating novel tools to assess psychological impairment in MS are urgently warranted. However, the present study presents several limitations in this regard. Indeed, the BOFRETTA 2.0 scale, which represents the core evaluation of the study, is not sufficiently explored, as the title of the manuscript would suggest. Following my major comments:

- Since the BOFRETTA 2.0 scale is not extensively used in the MS field and many neurologists may be unfamiliar with it, I would suggest the authors to describe more in detail the BOFRETTA 2.0 scale, providing (i) some examples of questions about death anxiety AND attitude towards death, (ii) describing in more detail how the sub-scores are obtained and (iii) reporting the normative values of this test. If, as I might understand, no normative values for this scale yet exist, I would suggest the authors to describe in more detail the mean-median-standard deviation-interquartile range of these scores reported by other authors in other populations. The reference 19, which is reported as normative control group, is in German and not easily accessible to international readers.

We thank the reviewer for this note which has led to a restructuring of the methods including an English language reference and some examples of the questions.

- The authors did not explore the impact of the MS phenotype on the numerous test scores reported in this study. I would suggest to report the test scores both in the whole patient population AND for RRMS and progressive MS separately. The authors should systematically explore between group differences in these two patient groups. Similarly, I suggest to explore correlations between the several NPS scores with disease duration, age and education and to explore the effect of sex.

We followed the reviewers advice and have reported for the test scores separately. We have also rearranged our tables, in order to clarify this point. We have also reported correlations on additional clinical and demographic variables, as far as they were available. 

- The authors should explicit how cognitive impairment was defined based on the Multiple Sclerosis Inventory of Cognition scores ( how "mild", "moderate", "heavy" cognitive impairment was defined?). The authors should as well report the cut-off values they used for HADS-A and HADS-D for defining "possible" anxiety/depression, presence and absence of anxiety/depression.

We thank the reviewer for this remark. We have now added the description of the MUSIC in the methods section and also added the cutoffs. We have also replaces the (wrong) word “heavy” by the word “severe”.

- Table 3. It is not clear to me which groups of patients were compared and why these group differences have not been explored systematically for all patient groups (depressed/not depressed, fatigue/no-fatigue, cognitive impaired/cognitively preserved, death anxiety-no death anxiety...). Moreover, group differences are not reported and discussed in the text. I suggest to reduce correlation analyses in the results section (bivariate and partial correlation analyses could be reported in a single table: authors can report only more meaningful results in the text thus making the reading more clear). 

We have now rearranged the tables according to the reviewers suggestions and have also reported the results in the result section.

- The authors report as a limitation of the study the absence of data on the years of education to correct for this confounder, but they report it in table 1.

We thank the reviewer for indicating this inconsistencies. We have now rephrased the discussion section appropriately.

- In the Discussion, the authors do not concentrate on death anxiety and attitude towards death assessed by BOFRETTA 2.0 (which should be the main topic of the paper) but rather discuss associations between depression/fatigue/cognitive impairment in MS patients, which are well established in the MS literature. The novelty of this work, which is to investigate death anxiety and to introduce a new NPS tool, is thus not sufficiently developed.

We are thankful for this comment which has led to a restructuration of the discussion in order to focus more on death and anxiety, relating them to the BOFRETTA.

- Abstract: the statement "It was shown that current psychopathology has a much stronger impact on the subject of death than the limitations caused by the neurological disease" is too declarative and it is not supported by the results of the study. Indeed EDSS is correlated with death anxiety. The term "current psychopathology" is too vague (what do the authors want to suggest?) .

We appreciate the reviewers statement concerning the term current psychopathology. Of course we have changed the expression into “actual psychopathology” which refers to the psychopathological status at the time of testing.

Round 2

Reviewer 2 Report

The current revised version of the manuscript has greatly improved.

I have no other suggestions.